# Fat-Containing Soft Tissue Tumors in Children, Adolescents, and Young Adults: Which Require Biopsy?

**DOI:** 10.3390/cancers15123228

**Published:** 2023-06-17

**Authors:** Liesbeth Cardoen, Nayla Nicolas, Violette Le Gaudu, Arnaud Gauthier, Matthieu Carton, Dominique Berrebi, Joanna Cyrta, Charlotte Collignon, Camille Cordero, Gaëlle Pierron, Stéphanie Pannier, Pascale Philippe-Chomette, Daniel Orbach, Hervé J. Brisse

**Affiliations:** 1Department of Imaging, Institut Curie, 75005 Paris, France; 2Department of Pathology, Institut Curie, 75005 Paris, France; 3Department of Biostatistics, Institut Curie, 75005 Paris, France; 4Department of Pathology, Assistance Publique des Hôpitaux de Paris, Hôpital Necker Enfants Malades, 75015 Paris, France; 5SIREDO Oncology Center (Care, Innovation and Research for Children and AYA with Cancer), Institut Curie, PSL University, 75005 Paris, France; 6Department of Somatic Genetics, Institut Curie, 75005 Paris, France; 7Paediatric Orthopaedic Service, Assistance Publique des Hôpitaux de Paris, Université Paris Cité, Hôpital Necker, 75015 Paris, France; 8Department of Pediatric Surgery, Assistance Publique des Hôpitaux de Paris, Hôpital Robert Debré, 75019 Paris, France

**Keywords:** soft tissue neoplasms, adipose tissue, pediatrics, biopsy, liposarcoma

## Abstract

**Simple Summary:**

Most fat-containing soft tissue tumors in children are benign, and liposarcomas are extremely rare in this age group. On the other hand, many benign fat-containing soft tissue tumors in children may demonstrate imaging features that can be reminiscent of adult liposarcoma. Although clinico-radiologic guidelines are available for adults, the diagnostic management of such tumors during childhood is still unclear. The aim of this study was to confirm the overall benignity of such tumors in a pediatric cohort and to highlight the clinical and imaging features that warrant a biopsy.

**Abstract:**

Purpose: To confirm the overall benignity of fat-containing soft tissue tumors (STT) on a pediatric cohort and to define the clinical and imaging features that warrant a biopsy. Methods: A retrospective monocentric study was conducted on patients aged less than 25 years consecutively referred for fat-containing STT to our Comprehensive Cancer Center between 1998 and 2022. Tumor imaging characteristics at diagnosis (US, CT, or MRI) were correlated with pathology. Results: The database extraction identified 63 fat-containing tumors with clinical, histologic, and imaging data available for review. In total, 58 (92%) were benign tumors: 36 lipoblastomas and lipomas, 12 fibrous hamartomas of infancy (FHI), 5 lipofibromatosis, 2 lipomas arborescens, 2 lipomatosis and 1 spindle-cell lipoma. Five patients (8%) were diagnosed with liposarcoma. Factors significantly correlated with malignancy were age >10 years old (*p* < 0.001), having a cancer-predisposing condition (*p* < 0.001), a percentage of fat <25% (*p* = 0.002), and a presence of myxoid zones (*p* < 0.001) on imaging. Conclusion: Most fat-containing STT in children may be classified as benign tumors based on clinics and imaging. The indication for biopsy could be limited to patients aged 10 years or more with either a cancer-predisposing condition or imaging features demonstrating either a low-fat component (<25%) or the presence of myxoid zones.

## 1. Introduction

The diagnosis of soft tissue tumors (STT) in children, adolescents, and young adults (AYA) is a common challenge. After clinical examination, the first-line imaging methods are usually conventional radiography and ultrasound with Doppler, which sometimes may be sufficient for diagnosis. MRI, due to its excellent tissue contrast, is the second-line technique and the main contributor to soft tissue analysis. However, biopsy still remains mandatory for definitive diagnosis in many cases [1,2].

In adults, although benign lipomas are very common, biopsy of fat-containing STT is frequently recommended—especially in large (>10 cm) or heterogeneous tumors—because liposarcoma is the most frequent malignant STT and well-differentiated liposarcoma may mimic mature lipoma upon MRI [3].

In children and AYA, fat-containing tumors account for only 6% of STT and include mainly benign tumors (i.e., lipoma/lipomatosis, lipoblastoma/lipoblastomatosis, hibernoma, fibrous hamartoma of infancy (FHI), involuting infantile and intramuscular hemangiomas, PTEN hamartoma, and fibro-adipose vascular anomaly (FAVA)) [4,5,6,7,8,9]. Liposarcomas are extremely rare in this age group and are often associated with a cancer-predisposing condition [10,11]. On the other hand, many benign fat-containing STT, such as lipoblastoma and FHI, demonstrate imaging features that may be reminiscent of adults’ liposarcoma, leading to unnecessary biopsy. In addition, biopsy in children is less straightforward compared to adults since it often requires general anesthesia. Therefore, the current diagnostic strategies used in adults’ fat-containing STT seem inadequate for children and AYA, and—to the best of our knowledge—no pediatric cohort has been published yet which could provide a rationale for biopsy indications within this specific age group.

The aim of this study was to retrospectively review the imaging features of a series of children and AYA consecutively referred to our cancer center for fat-containing STT and to identify clinico-radiologic features associated with malignancy in order to suggest an adjusted diagnostic strategy suited to this specific age group.

## 2. Materials and Methods

### 2.1. Ethics

This retrospective non-interventional study was approved by the institutional review board of our institution with a waiver of the requirement for patient consent.

### 2.2. Study Population

Our institutional database was queried based on the following inclusion criteria: patient age from 0 to 25 years; patient referred for a fat-containing STT located in the extremities, the trunk, or the head and neck region; and histopathologic confirmation of a fat-containing component. Non-inclusion criteria were fat-containing tumors (including germ cell tumors) located in either the mediastinum, abdomen, gonads, or central nervous system and incomplete or missing clinical, histopathologic, or radiologic data available for review.

Among the 870 patients referred for STT at our center between January 1998 and December 2020, our institutional database identified 87 eligible patients. After the exclusion of 24 incomplete medical records, 63 patients were enrolled in the study. 

### 2.3. Clinical, Histopathologic, and Imaging Data

Gender, age at diagnosis, medical history including personal and family history of cancer, tumor site, associated clinical symptoms (pain, skin changes), treatment, and follow-up data were collected from medical records. 

Histologic, immunohistochemical, and molecular data were obtained from reports of diagnostic biopsies (n = 30, including 21 percutaneous core-needle biopsies and 9 surgical biopsies) and/or surgical excision specimens (n = 57). Histopathology diagnostic categories followed the World Health Organization classification of soft tissue tumors [12]. Lipoblastomas and lipomas were grouped in a single category, as it is difficult to distinguish fully mature lipoblastoma from lipoma on standard histopathologic examination. Cytogenetics may help in differentiating the two entities since lipoblastomas show breakpoint abnormalities of the chromosomal region 8q11-13 or chromosome 8 polysomy, resulting in an upregulation of the PLAG1 gene product [6,8]. These alterations are not tested for in routine practice since the distinction is not clinically relevant due to both tumors being treated by conservative surgery.

MR scans were available for 56 patients, CT scans for 12, ultrasound for 42 and conventional radiography for 23. The images were analyzed in consensus by one junior and one senior pediatric radiologist (V.L. and L.C.).

The case report form for imaging data included tumor location (superficial or deep situated with respect to the superficial fascia), size (largest diameter), shape, margins, and echogenicity/density/signal compared to adjacent normal muscles. The percentage of fat within each tumor was visually and semi-quantitatively (i.e., 0%, <25%, 25–75%, >75%) assessed on CT and MR. Non-fatty components were also reported in terms of morphology (thin septa (<1 mm), thick septa (1–5 mm), nodules or patches) as in terms of echogenicity/density/signal and enhancement after contrast material injection. Myxoid areas were defined on MR images as areas with high signal (close to liquid) intensity on T2-WI and low signal (lower than muscles, close to water) intensity on T1-WI with enhancement after contrast injection. The adjacent invasion of neurovascular bundles or bone was also registered.

### 2.4. Statistical Analysis

Data were analyzed with descriptive statistics: numbers of cases and percentages for qualitative variables, and medians and ranges for continuous quantitative variables. Fisher exact and Wilcoxon tests were used for statistical comparison between the two groups: “benign tumors” and “malignant tumors”. *p*-values < 0.05 were considered statistically significant. Statistical analyses were performed using R software 4.1.1. 

## 3. Results

The study group included 63 tumors in 63 patients—19 females and 44 males, aged 0 to 23 years (median: 2.4 y). Histopathology identified 58/63 (92%) benign and 5/63 (8%) malignant tumors. Figure 1 shows the ratio distribution by histologic subtype according to age group. The clinical and imaging characteristics of each histologic subtype are summarized below and detailed in Table 1 and Table 2. In total, 30/63 (48%) patients had a diagnostic biopsy (21 percutaneous biopsies and 9 incisional biopsies). The majority of tumors (59/63 or 94%) were surgically resected. Figure 2 shows a summary diagram of the number of tumors that were biopsied at diagnosis and how they were handled.

A total of 30 tumors were biopsied at diagnosis, of which 21 had a percutaneous biopsy and 9 had an incisional biopsy. Of the tumors with percutaneous biopsy, 20/21 tumors were surgically excised, and 1 tumor was monitored. Of the tumors with incisional biopsy, 6/9 tumors were surgically excised, 2/9 tumors were monitored, and 1 tumor was treated with chemotherapy. Thirty-three tumors were operated on immediately without prior biopsy. 

### 3.1. Lipoblastoma and Lipoma

We observed 36 lipomas and lipoblastomas at various maturation stages, mainly in young children (median age 3 years). On MRI or CT, all tumors contained ≥25% fat, and 29 cases (83%) contained >75% fat. The non-fatty component had variable imaging features, including thin or thick septa or nodular or more patchy areas, frequently showing contrast enhancement (Figure 3). In four patients, fluorescence in situ hybridization (FISH) demonstrated PLAG1 rearrangements confirming the diagnosis of lipoblastoma. Twelve were first biopsied for diagnostic confirmation and all tumors were treated with conservative surgical resection. One intramuscular lipoblastoma located in the right gluteal region with extension into the greater sciatic notch recurred 21 months after the initial surgery and was cured by a second complete surgery. 

### 3.2. Fibrous Hamartoma of Infancy

All 12 cases of FHI occurred in patients under 1 year of age at diagnosis (median 3 months). The most common anatomical locations were the trunk and upper limbs. The ultrasonographic appearance was characteristic, showing a heterogeneous echostructure with interlaced hypoechoic and hyperechoic bands. On MRI, FHI were soft tissue masses with ill-defined margins interspersed with fat lobules (Figure 4). They contained a relatively low amount of fat (<25% in 9/12 cases) and nodular or patchy non-fatty components with intermediate signal intensity on T2-weighted images and contrast enhancement. Half of the patients were first biopsied for diagnostic confirmation, and surgical excision was further decided upon for 10 patients without recurrence (only clinical follow-up for 2 patients). 

### 3.3. Lipofibromatosis

Lipofibromatosis occurred in one female and four male patients, aged 0 to 8 years (median: 2 months). MRI showed lipomatous lesions with more than 75% fat (4/5) and different appearances of the non-fatty components varying from thick septa to nodular-shaped solid components. Three were first biopsied for diagnostic confirmation. The treatment varied from simple monitoring to chemotherapy to surgical excision. 

### 3.4. Other Benign Tumors

Imaging for the patients with lipoma arborescens (n = 2), lipomatosis (n = 1), lipomatosis of the nerve (n = 1) and spindle cell lipoma (n = 1) showed masses composed predominantly of fat. Clinical and other imaging features are detailed in Table 1 and Table 2. All, except the case of lipomatosis, had a core needle biopsy before surgical excision.

### 3.5. Liposarcoma

Liposarcoma occurred in two female and three male patients aged 12 to 20 years (median 14 years). Masses were located in the thigh (3/5), the orbit (1/5), and the breast (1/5). Three tumors were grade 1 myxoid liposarcomas (all with FUS::DDIT3 fusion transcript), and two tumors were grade 2 pleomorphic liposarcomas. All tumors were localized at diagnosis. Three patients had a predisposing condition: a 14-year-old girl with myxoid liposarcoma of the thigh underwent radiotherapy 6 years earlier for pelvic Ewing’s sarcoma; a 12-year-old boy and an 18-year-old girl with pleomorphic liposarcoma of the orbit and the breast, respectively, were diagnosed with Li–Fraumeni syndrome. MRI performed in four cases revealed less than 25% fat (2/4) or no fat at all (2/4) with nodular or patchy non-fatty components with contrast enhancement and myxoid areas (4/4) (Figure 5). All patients were first biopsied and further treated with surgical excision (R0), preceded by neoadjuvant chemotherapy in two cases. The boy with orbital liposarcoma relapsed 16 months after exenteration, leading to new surgery and radiotherapy. Unfortunately, he progressed locally and developed bone metastases, and the patient died 1 year after the recurrence. The girl with breast pleomorphic liposarcoma did not relapse but developed another sarcomatous tumor in the contralateral breast. She and all patients with a tumor located in the thigh are currently in first complete remission, with a median follow-up of 6 years (range 4–10 years). 

### 3.6. Liposarcoma vs. Benign Tumor 

Univariate analysis of all clinical and imaging characteristics was performed to screen for predictors of malignancy (Table 3). Variables significantly related to tumor malignancy were: predisposing condition (genetic predisposition or previous oncologic treatment), age at diagnosis (>10-year-old), low fat percentage (<25%), and presence of a myxoid component. These items were strongly associated with malignancy with a negative predictive value of 100% (95CI: 93–100%) for age at diagnosis >10 years old, fat percentage <25%, the presence of a myxoid component and a positive predictive value of 100% (95CI: 29–100%) for predisposition (Table 4). The complete separation of data was observed between benign and malignant groups for the parameters of age, percentage of fat, and myxoid components. Multivariate logistic regression was then impossible to perform.

In Figure 6, we suggest a decision tree to be used in routine practice for discriminating benign versus malignant fat-containing STTs in the pediatric and AYA populations.

## 4. Discussion

This single-center series of children and AYA with fat-containing STT confirms that most tumors are benign within this age group and that liposarcomas are extremely rare in children and occur mostly in adolescents, in agreement with the literature [5,10,11,13,14,15].

Therefore, the current diagnostic strategy for fat-containing STT of adults should not be applied in children and AYA since this would lead to unnecessary diagnostic biopsies and general anesthesia. This series proposes a diagnostic strategy adjusted for children and AYA based on both clinical and radiological criteria.

Among the clinical criteria, our results demonstrate the major criterion of age at diagnosis. All patients diagnosed with liposarcoma in this series were 12 years or older. In the liposarcoma series published by Huh et al. [10], all 33 patients were older than 11 years, and Alaggio et al. [11] reported only 4 of 81 patients with liposarcoma younger than 11 years. A cancer-predisposing condition was also observed in three of our five patients with liposarcoma. Both tumors associated with Li–Fraumeni syndrome were pleomorphic liposarcomas, a subtype known to be more aggressive and associated with a poorer prognosis than other types of liposarcoma. Our three other cases of liposarcoma were of the myxoid type, i.e., the most common type described in children [10,11,16]. Although usually considered suspicious clinical signs in adults, local pain and large tumor size were not associated with malignancy in this cohort (and were absent in all our liposarcoma cases).

Imaging criteria allowing the identification of malignant fat-containing tumors in adults (i.e., size over 10 cm, association of fatty and non-fatty components, solid nodular areas or thick septa) [3] cannot be used in children since lipoblastomas and FHI, the most common tumors encountered in children, demonstrate features which would be interpreted in adults as suggestive for malignancy. On the other hand, since low-grade well-differentiated liposarcoma in adults may mimic benign lipoma, it is recommended that a preoperative biopsy be performed for large (>10 cm) or deep-situated tumors (risk of incomplete surgery), even for totally fatty tumors [3]. Our results demonstrate that totally fatty tumors or tumors with >75% fat are always benign in children, even when large or deep-situated. Hence, the imaging criteria used to decide diagnostic biopsy in adults definitely cannot be applied to the pediatric population.

Among the radiological criteria, a low amount of fat (<25%) and myxoid areas were observed in all our liposarcoma cases, these findings being widely described in the literature regarding adults [9], but they were not observed in any of the benign tumors of our cohort occurring in patients older than 10 years (but were observed in three lipoblastomas in much younger patients). Therefore, these two criteria should be also considered as reliable signs of malignancy in the AYA population.

The proposed decision tree for discriminating benign versus malignant fatty tumors is based on both clinical and radiological criteria. If this strategy had been used in our cohort, all five liposarcoma patients would still have benefited from biopsy, while 83% (25/30) of diagnostic biopsies performed in our cohort (without modification of the treatment strategy) would have been avoided. 

Lipoblastomas and lipomas were the most frequent fat-containing STT in children in our study. Previously published reports separated these entities, but all considered them to be the two most common fat-containing STT in children [4,6,7,8]. Although lipoblastomas are principally tumors of infancy and early childhood, they can occur in older children, adolescents, and young adults. 

All lipoblastomas and lipomas showed a large predominance of fat. Although some articles reported data on lipoblastomas with very little adipose tissue to no adipose tissue at all, especially in very young patients [4,7,17], this finding was not confirmed in our series. The non-fatty component features of lipoblastomas included thin or thick septa-like, nodular, or patchy areas frequently showing contrast enhancement. This heterogeneous aspect of MR imaging is consistent with previously published data [4,7,8,18,19] and is explained by the histologic composition including variable proportions of adipocytes at various maturation stages, myxoid stroma, and fibrous septa [6]. Myxoid components may occur in lipoblastomas and therefore should not be considered a pejorative sign unless the age is higher than 10 years. Scarce cases of predominantly myxoid undifferentiated lipoblastomas (some associated with specific PLAG1-HAS2 fusion transcript) [20,21,22,23] have been reported in children. In such cases, the diagnosis of lipoblastoma cannot be suggested by imaging, and diagnostic biopsy should be recommended as other benign and malignant myxoid tumors may occur.

In agreement with published data [24,25,26], all FHI of this series occurred within the first year of life, were located in the upper extremity or the trunk, and showed the characteristic ultrasound “serpentine pattern” [25] and equivalent MR features mirroring the histologic findings of fatty, fibrous, and mesenchymal trabecular elements [26,27]. This clinico-radiological pattern should be recognized and should not warrant any diagnostic biopsy. 

Lipomatosis is usually seen in young children less than 2 years of age [4,7,9,17]. Lipomatous overgrowth can be associated with different congenital limb overgrowth syndromes. Mutations at various points of the PI3K/AKT/mTOR cell-signaling pathway result in diverse disorders characterized by limb overgrowth and vascular anomalies with substantial phenotypic overlap [28].

Lipofibromatosis is a rare entity first described in 2000 [29], and current data suggest that the various imaging aspects reflect the various proportions of fat and fibroblastic tissue [30,31]. The five cases of our series had very different imaging aspects but occurred before the age of 8 years and did not contain myxoid areas.

Other less frequent histologic subtypes of fat-containing STT, such as hibernoma, are rare in children [6,8,17,32]. Intramuscular capillary-type hemangioma, PTEN hamartoma of soft tissue (PHOST) and fibro adipose vascular anomaly (FAVA) were also not observed in this series. In the published series of Yilmaz et al. [33], adipose tissue was present in histopathologic results in all cases of intramuscular capillary-type hemangioma. However, the fat content on imaging was much lower than would be expected with a lipomatous tumor. No reliable imaging criteria were found to differentiate intramuscular capillary-type hemangioma from soft-tissue sarcomas such as rhabdomyosarcoma or liposarcoma. Therefore, the authors recommend a biopsy in all cases. The PTEN hamartoma syndrome family is a spectrum of disorders with mutations in the PTEN gene, including PTEN hamartomas. The lesion typically includes infiltrative solid fatty and fibrous muscular portions with strong enhancement associated with tortuous intralesional veins and thick-walled arteries. The vascular channels can be slow or fast with arteriovenous shunting. FAVA is a more recently described rare complex vascular malformation most commonly caused by somatic mutations in the PIK3CA gene. Fibro-fatty infiltration, intralesional intertwining dilated veins, subcutaneous phlebectasia, and fatty overgrowth are hallmark MRI features [34]. 

The limitations of this study include the relatively small sample size, which limits statistical power. Only parameters very strongly associated with malignancy have been highlighted, and more modest links might have been occulted. The monocentric recruitment in a cancer center, as well as the inclusion of only those tumors which were either biopsied or removed, induces a recruitment bias leading to an overestimation of the malignancy rate. However, despite this bias, the relative prevalence of benign tumors was still high, enhancing the relevance of our recommendation. Finally, our results obtained from post hoc analysis on small samples are exploratory and will require further tests on a validation cohort.

## 5. Conclusions

In conclusion, fat-containing STT of the extremities, the trunk, or the head and neck in children and AYA are almost exclusively benign and dominated by lipoblastomas at various stages of maturation. Our study highlights several clinical and imaging features associated with malignancy that should warrant a biopsy: age ≥10 years old, predisposition (genetic predisposition to cancer or previous oncologic treatment), a low amount of fat (<25%), and presence of myxoid areas. In other cases, a biopsy could be avoided, knowing that predominantly fatty tumors are all benign in children even when containing thick septa or nodular non-fatty components. Based on this rationale, we propose a minimally invasive diagnostic strategy approach more suited to children, adolescents, and young adults. The proposed strategy must be statistically confirmed in a large multi-center study.

## Figures and Tables

**Figure 1 cancers-15-03228-f001:**
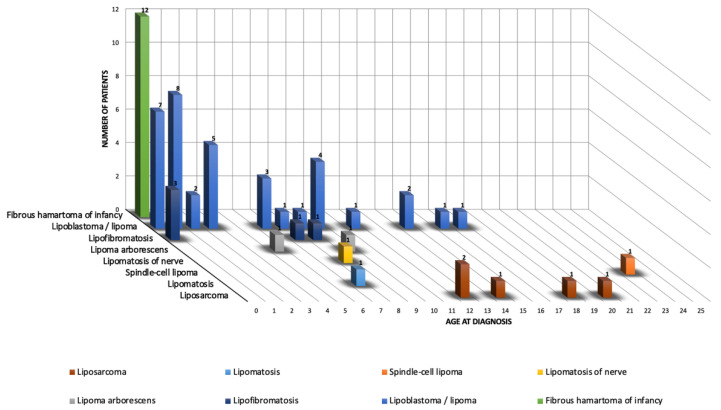
Age distribution according to histologic subtype. The horizontal axis represents the age at diagnosis in years. The depth axis represents the histological diagnosis. The vertical axis represents the number of cases.

**Figure 2 cancers-15-03228-f002:**
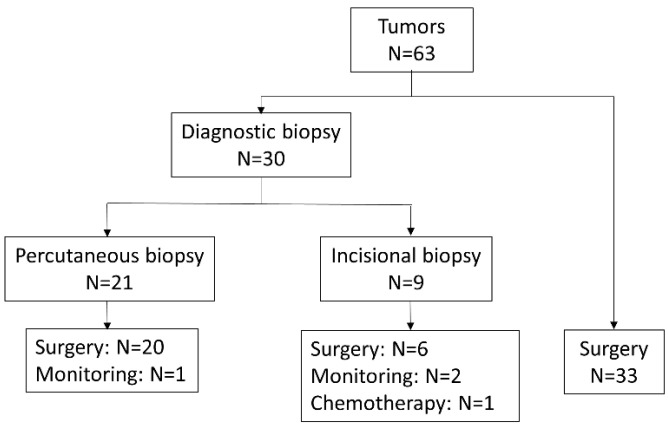
Summary diagram of number of diagnostic biopsies and handling of the tumors.

**Figure 3 cancers-15-03228-f003:**
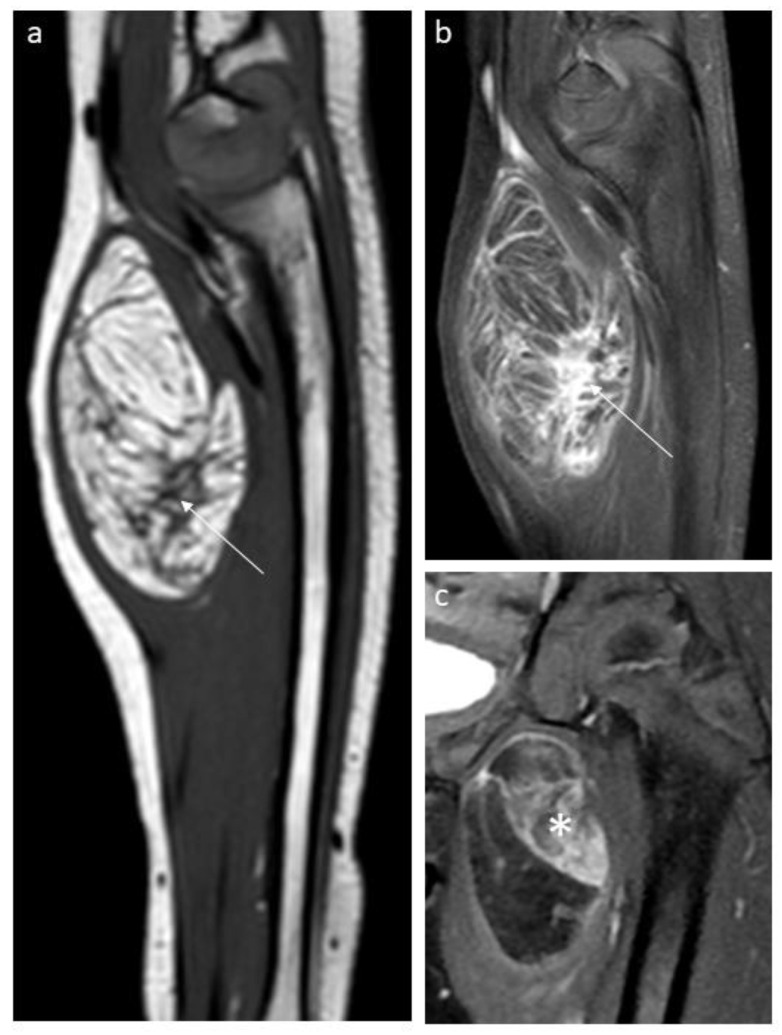
Aspects of non-fatty components in lipoblastomas and lipomas. Sagittal T1-weighted (**a**) and sagittal contrast-enhanced T1-weighted (**b**) MR images of a forearm lipoblastoma in a 3-year-old boy demonstrating thick septa (arrow). (**c**) Coronal T2-weighted MR image of a lipoblastoma located in the thigh in a 3-year-old girl with patchy, enhancing non-fatty component (star).

**Figure 4 cancers-15-03228-f004:**
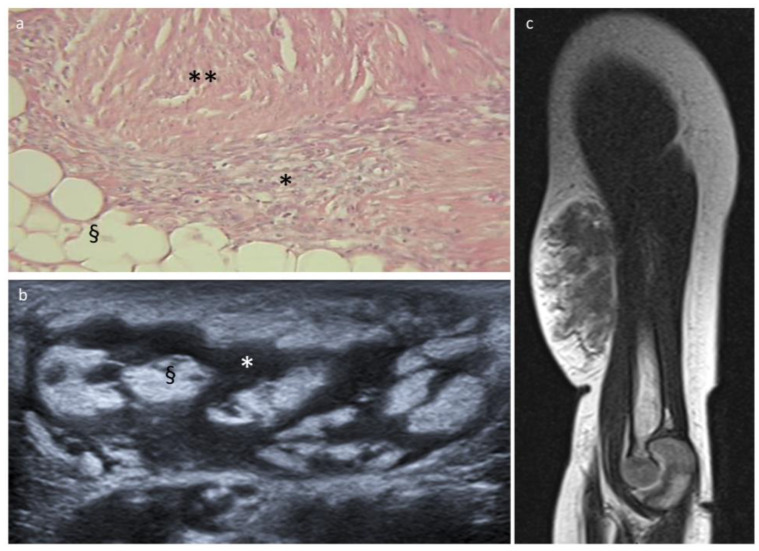
Fibrous hamartoma of infancy and histopathologic correlation. (**a**) Histologic section with hematoxylin and eosin stain shows the three characteristic components: adipose tissue (§), fibrous tissue (**), and immature mesenchymal cells (*). (**b**) US image shows interlaced hyper- and hypoechoic bands reflecting the histology: hyperechoic fat (§) and hypoechoic fibrous/mesenchymal areas (*). (**c**) MRI sagittal T2-weighted sequence shows an FHI located at the arm of an 8-month-old boy with ill-defined margins and a relatively low amount of fat.

**Figure 5 cancers-15-03228-f005:**
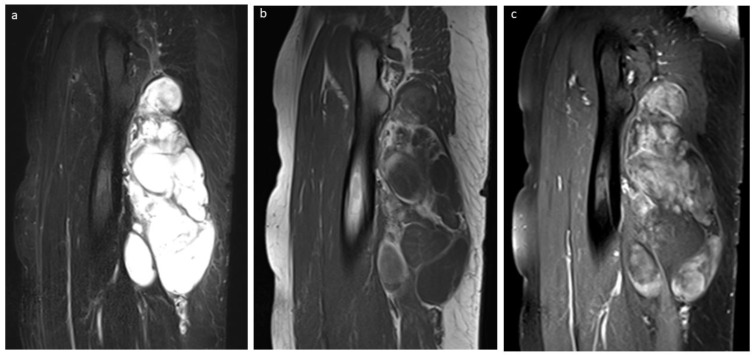
Myxoid liposarcoma in a 20-year-old boy. (**a**) Sagittal STIR image shows a hyperintense, large, lobated mass located in the thigh. (**b**) Sagittal T1-weighted image shows the absence of adipose tissue, and (**c**) sagittal contrast-enhanced T1-weighted image demonstrates the contrast enhancement of the myxoid areas.

**Figure 6 cancers-15-03228-f006:**
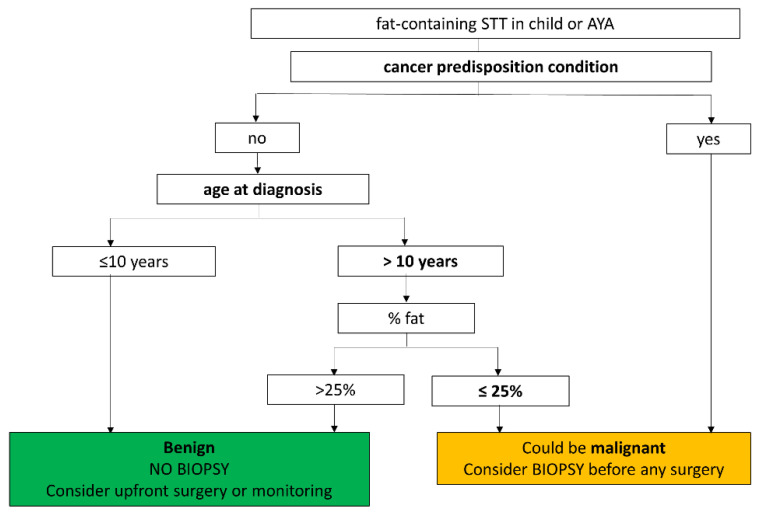
Decision tree for discriminating benign versus malignant fat-containing STT in the pediatric and AYA populations.

**Table 1 cancers-15-03228-t001:** Clinical features for each tumor subtype.

	Lipoblastoma/Lipoma	FHI	LF	Lipoma Arborescens	Lipomatosis	Lipomatosis of Nerve	Spindle Cell Lipoma	Liposarcoma
Number of cases	36	12	5	2	1	1	1	5
Gender female/male	11/25	3/9	1/4	1/1	0/1	0/1	1/0	2/3
Age: median [range]	3 yo [0–17 yo]	3 mo [0–10 mo]	2 mo [0–8 yo]	7 yo [5–9 yo]	7 yo	8 yo	23 yo	14 yo [12–20 yo]
Predisposition	0	0	0	0	0	0	0	3
Pain	5	1	0	1	0	0	0	0
Skin changes	2	4	1	1	0	0	0	1
Location								
Head and neck	2	1	-	-	-	-	-	1
Trunk	17	5	-	-	-	-	1	1
Upper extremity	7	5	1	-	-	-	-	-
Lower extremity	8	1	4	1	1	1	-	3
Scrotum	2	-	-	-	-	-	-	-

yo = years old, mo = months old, FHI = fibrous hamartoma of infancy, LF = lipofibromatosis. Values represent number of cases unless otherwise specified.

**Table 2 cancers-15-03228-t002:** Imaging features for each tumor subtype.

	Lipoblastoma/Lipoma	FHI	LF	Lipoma Arborescens	Lipomatosis	Lipomatosis of Nerve	Spindle Cell Lipoma	Liposarcoma
**Depth**								
Superficial	13	9	2	-	1	1	1	1
Deep	23	3	3	2	-	-	-	4
**Size (mm):** mean [range]	56 [15–170]	39 [25–65]	66 [15–82]	92.5 [65–120]	45	75	75	71 [15–215]
**Margins**								
Well-defined	33	5	1	2	-	1	1	3
Ill-defined	3	7	4	-	1	-	-	2
**Shape**								
Ovoid	22	10	2	-	-	1	-	2
Polylobate	14	2	-	2	-	-	1	3
Diffuse	-	-	3	-	1	-	-	-
**Echostructure**				-	-			
Homogeneous	9 (/23)	-	-			-	1	-
Heterogeneous	14	10 (/10)	2 (/2)			1	-	3 (/3)
**Echogenicity**				-	-			
Isoechoic	1 (/23)	-	-			-	-	-
Hyperechoic	16	-	2 (/2)			1	1	3 (/3)
Hypoechoic	2	-	-			-	-	-
Mixed	4	10 (/10)	-			-	-	-
**% of fat**								
0%	-	-	0	-	-	-	-	2 (/4)
<25%	-	9	1	-	-	-	-	2
25–75%	6 (/35)	1	0	1	-	-	-	-
>75%	29	2	4	1	1	1	1	-
**Non-fatty components (NFC)**								
None	2 (/35)	-	2	-	1	-	-	-
Thin septa	16	-	-	-	-	1	-	-
Thick septa	7	-	2	1	-	-	-	-
Nodules	5	2	-	-	-	-	-	1 (/4)
Patches	5	10	1	1	-	-	1	3
**NFC: T1-weighted signal**					-			
Isosignal	8 (/29)	9	3 (/3)	1		-	1	3 (/4)
Hyposignal	20	1	-	1		1	-	1
Hypersignal	1	2	-	-		-	-	-
**NFC: T2-weighted signal**					-			
Isosignal	1 (/28)	-	-	-		-	1	-
Hyposignal	19	1	1 (/3)	1		1	-	-
Hypersignal	4	-	-	1		-	-	4 (/4)
Intermediate	4	11	2	-		-	-	-
**Enhancement**	21 (/31)	10 (/11)	3	2	-	-	1	4 (/4)
**Myxoid zone**	3 (/28)	-	-	-	-	-	-	4 (/4)
**Neurovascular encasement**	4	-	3	-	1	1	-	1
**Bone involvement**	-	-	-	-	-	-	-	-
**Calcifications**	-	-	-	-	-	-	-	-
**Cystic zone**	1	-	-	1	-	-	-	-

NFC = non-fatty components, FHI = fibrous hamartoma of infancy, LF = lipofibromatosis. Values represent number of cases unless otherwise specified. Some data could not be evaluated in all patients (missing MRI or contrast material injection): the number of patients in which they were evaluated are specified in brackets.

**Table 3 cancers-15-03228-t003:** Comparison of clinical and radiologic variables between the groups “malignant tumors” and “benign tumors”.

Factor		Benign	Malignant	Total	Fisher Test
Sex	Female	17 (29.3%)	2 (40%)	19 (30.2%)	*p* = 0.63
	Male	41 (70.7%)	3 (60%)	44 (69.8%)	
Predisposition	Yes No	0 (0%) 58 (100%)	3 (60%) 2 (40%)	3 (4.8%) 60 (95.2%)	*p* < 0.001
Age at diagnosis	≤10 y.o. >10 y.o.	52 (89.7%) 6 (10.3%)	0 (0%) 5 (100%)	52 (82.5%) 11 (17.5%)	*p* < 0.001
Location	Upper extremity	13 (22.4%)	0 (0%)	13 (20.6%)	*p* = 0.26
	Lower extremity	17 (29.3%)	3 (60%)	20 (31.7%)	
	Trunk	23 (39.7%)	1 (20%)	24 (38.1%)	
	Head and neck	3 (5.2%)	1 (20%)	4 (6.3%)	
	Other	2 (3.4%)	0 (0%)	2 (3.2%)	
Pain	Yes	7 (12.7%)	0 (0%)	7 (11.7%)	*p* = 1
	No	48 (87.3%)	5 (100%)	53 (88.3%)	
	NA	3	0	3	
Skin changes	No	46 (83.6%)	4 (80%)	50 (83.3%)	*p* = 1
	Yes	9 (16.4%)	1 (20%)	10 (16.7%)	
	NA	3	0	3	
Depth	Supra-aponeurotic	27 (46.6%)	1 (20%)	28 (44.4%)	*p* = 0.5
	Sub-aponeurotic	31 (53.4%)	4 (80%)	35 (55.6%)	
Shape	Ovoid or round	35 (60.3%)	2 (40%)	37 (58.7%)	*p* = 0.53
	Polylobed	19 (32.8%)	3 (60%)	22 (34.9%)	
	Diffuse	4 (6.9%)	0 (0%)	4 (6.3%)	
Margins	Well-defined	43 (74.1%)	3 (60%)	46 (73%)	*p* = 0.65
	Ill-defined	13 (22.4%)	2 (40%)	15 (23.8%)	
	ND	2 (3.4%)	0 (0%)	2 (3.2%)	
Echostructure	Homogeneous	10 (27%)	0 (0%)	10 (25%)	*p* = 0.56
	Heterogeneous	27 (73%)	3 (100%)	30 (75%)	
	NA	21	2	23	
Echogenicity	Iso-echoic	1 (2.7%)	0 (0%)	1 (2.5%)	*p* = 0.41
	Hyper-echoic	20 (54.1%)	3 (100%)	23 (57.5%)	
	Hypo-echoic	2 (5.4%)	0 (0%)	2 (5%)	
	Mixt hyper and hypo-echoic	14 (37.8%)	0 (0%)	14 (35%)	
	NA	21	2	23	
Percentage of fat	0%	0 (0%)	2 (50%)	2 (3.2%)	*p* < 0.001
	<25%	10 (17.2%)	2 (50%)	12 (19.4%)	
	25–75%	8 (13.8%)	0 (0%)	8 (12.9%)	
	>75%	40 (69%)	0 (0%)	40 (64.5%)	
	NA	0	1	1	
Non-fatty components	None	5 (8.8%)	0 (0%)	5 (8.2%)	*p* = 0.38
	Thin septa < 1 mm	17 (29.8%)	0 (0%)	17 (27.9%)	
	Thick septa 1–5 mm	10 (17.5%)	0 (0%)	10 (16.4%)	
	Nodules	7 (12.3%)	1 (25%)	8 (13.1%)	
	Patches	18 (31.6%)	3 (75%)	21 (34.4%)	
	NA	1	1	2	
NFC: T1-weighted signal	Isosignal	22 (45.8%)	3 (75%)	25 (48.1%)	*p* = 0.68
	Hyposignal	23 (47.9%)	1 (25%)	24 (46.2%)	
	Hypersignal	3 (6.2%)	0 (0%)	3 (5.8%)	
	NA	10	1	11	
NFC: T2-weighted signal	Iso T2	2 (4.3%)	0 (0%)	2 (3.9%)	*p* = 0.001
	Hyposignal	23 (48.9%)	0 (0%)	23 (45.1%)	
	Hypersignal	5 (10.6%)	4 (100%)	9 (17.6%)	
	Intermediate signal	17 (36.2%)	0 (0%)	17 (33.3%)	
	NA	11	1	12	
Enhancement	Yes	37 (72.5%)	4 (100%)	41 (74.5%)	*p* = 0.56
	No	14 (27.5%)	0 (0%)	14 (25.5%)	
	NA	7	1	8	
Cystic zones	Yes	2 (3.6%)	0 (0%)	2 (3.4%)	*p* = 1
	No	54 (96.4%)	3 (100%)	57 (96.6%)	
	NA	2	2	4	
Myxoid zones	No	50 (94.3%)	0 (0%)	50 (87.7%)	*p* < 0.001
	Yes	3 (5.7%)	4 (100%)	7 (12.3%)	
	NA	5	1	6	
Neurovascular encasement	Yes	9 (16.1%)	1 (25%)	10 (16.7%)	*p* = 0.53
	No	47 (83.9%)	3 (75%)	50 (83.3%)	
	NA	2	1	3	

*p*-values < 0.05 were considered statistically significant. NA: not available (missing data).

**Table 4 cancers-15-03228-t004:** Results of the diagnostic values (malignant/benign) of the different variables.

Factors	Age at Diagnosis (>10 Years Old)	Cancer Predisposition	% Fat	Myxoid Component
N patients	63	63	62	57
% of population	100.0%	100.0%	98.4%	90.5%
true positive	5	3	4	4
false negative	0	2	0	0
true negative	52	58	48	50
false positive	6	0	10	3
apparent prevalence	17.5% (9.1–29.1%)	4.8% (1.0–13.3%)	22.6% (12.9–35.0%)	12.3% (5.1–23.7%)
true prevalence	7.9% (2.6–17.6%)	7.9% (2.6–17.6%)	6.5% (1.8–15.7%)	7.0% (1.9–17.0%)
sensitivity	100.0% (47.8–100.0%)	60.0% (14.7–94.7%)	100.0% (39.8–100.0%)	100.0% (39.8–100.0%)
specificity	89.7% (78.8–96.1%)	100.0% (93.8–100.0%)	82.8% (70.6–91.4%)	94.3% (84.3–98.8%)
positive predictive value	45.5% (16.7–76.6%)	100.0% (29.2–100.0%)	28.6% (8.4–58.1%)	57.1% (18.4–90.1%)
negative predictive value	100.0% (93.2–100.0%)	96.7% (88.5–99.6%)	100.0% (92.6–100.0%)	100.0% (92.9–100.0%)

The 95% confidence intervals are given in brackets. The percentage (%) of fat was only evaluable on cross-sectional imaging, available in 4/5 malignant tumors. The presence of myxoid areas was only evaluable on contrast-enhanced MRI, available in 4/5 malignant and 53/58 benign tumors.

## Data Availability

The data presented in this study are available on request from the corresponding author.

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
