# Peer review of "Fat-Containing Soft Tissue Tumors in Children, Adolescents, and Young Adults: Which Require Biopsy?"

_cancers, 2023, doi:10.3390/cancers15123228_

Round 1

Reviewer 1 Report

I was glad to review the work of the authors regarding this very interesting article on Fat-Containing Soft-Tissue Tumors in Children, Adolescents, and Young Adults. The manuscript is well-written and the incorporated tables and figures make the study easy to follow.

The topic is very interesting. There are not many articles published which discuss the role fat containing issue and its correlation with biopsy requirement. 

The negative point of this manuscript is the low number of patients and poor statistical analysis. Another negative point is that the study was done retrospectively as well as it is single center experience.

It would be better to have more data from different centers in order to make better statistic analysis as well as have clear results.

The positive points of this manuscript is that there are not many similar articles published in the literature. 

Authors should claim the limitations of their study.

In my opinion it can be accepted for publication since it is a unique topic and scientifically well written.

I strongly recommend acceptance for publication of the paper after minor changes.

Despite the major advances in clinical-radiologic features associated with this malignancy, there are still numerous unanswered questions regarding the adjusted diagnostic strategy suited to this specific age group.

I would like a brief discussion on the spindle cell variant of embryonal rhabdomyosarcoma.

 Please consider citing the recently published article:

https://pubmed.ncbi.nlm.nih.gov/32801444/

How many of your patients had this rare subtype?

Reviewer 2 Report

I  have read this study with great interest, However, it raised my several concerns:

1. I am not convinced that upper limit of 25 is correct, we can several well differentiated liposarcomas especially retroperitoneal sarcomas at this age, and sometimes the only location, radiological imaging and presence of mdm2 confirmed with the biopsy confirm the diagnosis, and I do not think that discontinuation biopsy before possible large surgery is proper management.

2.  the numbers are very low and they may be biased especially in young adults population, I would suggest to perform multicenter study including adult sarcomas centers if age 25 is chosen.

Round 2

Reviewer 2 Report

thank you for improvement of this version, I am in favor to approve it